# Protein Farnesylation on Nasopharyngeal Carcinoma, Molecular Background and Its Potential as a Therapeutic Target

**DOI:** 10.3390/cancers14122826

**Published:** 2022-06-08

**Authors:** Eiji Kobayashi, Satoru Kondo, Hirotomo Dochi, Makiko Moriyama-Kita, Nobuyuki Hirai, Takeshi Komori, Takayoshi Ueno, Yosuke Nakanishi, Miyako Hatano, Kazuhira Endo, Hisashi Sugimoto, Naohiro Wakisaka, Tomokazu Yoshizaki

**Affiliations:** Department of Otolaryngology—Head and Neck Surgery, Kanazawa University, 13-1 Takaramachi, Kanazawa 920-8640, Japan; e_kobayashi@med.kanazawa-u.ac.jp (E.K.); ksatoru@med.kanazawa-u.ac.jp (S.K.); h_dochi@med.kanazawa-u.ac.jp (H.D.); mkita@med.kanazawa-u.ac.jp (M.M.-K.); nhhira@med.kanazawa-u.ac.jp (N.H.); takkomori@med.kanazawa-u.ac.jp (T.K.); uenotaka@med.kanazawa-u.ac.jp (T.U.); nakanish@med.kanazawa-u.ac.jp (Y.N.); mhatano@med.kanazawa-u.ac.jp (M.H.); endok@med.kanazawa-u.ac.jp (K.E.); sugimohi@med.kanazawa-u.ac.jp (H.S.); wakisaka@med.kanazawa-u.ac.jp (N.W.)

**Keywords:** nasopharyngeal carcinoma, Epstein–Barr virus, farnesylation, farnesyltransferase inhibitor, RAS

## Abstract

**Simple Summary:**

Nasopharyngeal carcinoma is distinguished from other head and neck carcinomas by the association of its carcinogenesis with the Epstein–Barr virus. It is highly metastatic, and a novel therapeutic modality for metastatic nasopharyngeal carcinoma is keenly awaited. Protein farnesylation is a C-terminal lipid modification of proteins and was initially investigated as a key process in activating the RAS oncoprotein through its association with the cellular membrane structure. Since then, more and more evidence has accumulated to indicate that proteins other than RAS are also farnesylated and have significant roles in carcinogenesis. This review delineates molecular pathogenesis through protein farnesylation in the context of nasopharyngeal carcinoma and discusses the potential of farnesylation as a therapeutic target.

**Abstract:**

Nasopharyngeal carcinoma (NPC) is one of the Epstein–Barr virus (EBV)-associated malignancies. NPC is highly metastatic compared to other head and neck carcinomas, and evidence has shown that the metastatic features of NPC are involved in EBV infection. The prognosis of advanced cases, especially those with distant metastasis, is still poor despite advancements in molecular research and its application to clinical settings. Thus, further advancement in basic and clinical research that may lead to novel therapeutic modalities is needed. Farnesylation is a lipid modification in the C-terminus of proteins. It enables proteins to attach to the lipid bilayer structure of cellular membranes. Farnesylation was initially identified as a key process of membrane association and activation of the RAS oncoprotein. Farnesylation is thus expected to be an ideal therapeutic target in anti-RAS therapy. Additionally, more and more molecular evidence has been reported, showing that proteins other than RAS are also farnesylated and have significant roles in cancer progression. However, although several clinical trials have been conducted in cancers with high rates of *ras* gene mutation, such as pancreatic carcinomas, the results were less favorable than anticipated. In contrast, favorable outcomes were reported in the results of a phase II trial on head and neck carcinoma. In this review, we provide an overview of the molecular pathogenesis of NPC in terms of the process of farnesylation and discuss the potential of anti-farnesylation therapy in the treatment of NPC.

## 1. Introduction

Nasopharyngeal carcinoma (NPC) is endemic in the southern part of China and Southeast Asia. In contrast, its incidence is rather low in Western countries, as well as in Japan [1]. NPC is a highly metastatic malignancy compared with other head and neck carcinomas. The most common initial symptom of NPC is a neck mass, which results from cervical lymph node metastasis [2,3]. Therefore, systemic chemotherapy is a key modality in the treatment of NPC. However, NPC differs greatly from other head and neck carcinomas in its carcinogenesis. Epstein–Barr virus (EBV) infection has been characterized as a trigger for carcinogenesis in virtually all cases of NPC, regardless of whether it is endemic or non-endemic for the regions. EBV persists in NPC tissue predominantly as a latent infection termed latency type Ⅱ, where its expression is restricted to certain viral genes: latent membrane protein 1 (LMP1), LMP2, nuclear antigen 1 (EBNA1), EBV-encoded RNAs (EBERs), and microRNAs (miRNAs) encoded in the BamHⅠ rightward transcript (BART) region [2,3,4,5].

Of those protein products of viral genes, LMP1 is a primary oncoprotein of EBV and plays an important role in the carcinogenesis of NPC [6]. The protein consists of 386 amino acids that self-associate in the plasma membrane. It contains the structure of a terminal cytoplasmic end, transmembrane region with six hydrophobic domains, and two main signaling domains in the carboxyl-terminus called carboxyl-terminal activating regions 1 and 2 (CTAR1 and 2) [7]. LMP1 acts independently of a ligand as a constitutively active tumor necrosis factor (TNF) receptor. The signaling is mediated by binding of CTAR1 or CTAR2 with adaptor molecules. CTAR1 binds TNF-receptor-associated factors (TRAFs), and CTAR2 binds the TNF-receptor-associated death domain or the receptor-interacting protein (RIP) [8,9,10]. LMP1 activates several signaling pathways known in cancer progression, including the NF-κB [11,12], mitogen-activated protein kinase (MAPK) [13], c-Jun N-terminal kinase (JNK) [14], and phosphatidylinositol 3-kinase (PI3K)-Akt pathways [15]. Additionally, cellular markers associated with G_1_/S cell cycle transition are mediated by LMP1 [16]. These properties of LMP1 account for EBV’s properties of transformation and oncogenesis. For example, LMP1 activates several oncoproteins, such as MMP9, IL-8, FGF-2, Twist, and Cox-2, through the NF-κB pathway [17,18,19,20,21].

Recent global epidemiological analysis showed that the mortality rate of NPC cases has decreased and the survival rate has increased, probably due to advancements in diagnostic techniques, chemotherapy, and radiotherapy [22,23], but the prognosis of advanced cases, especially those with distant metastasis, is still poor [24]. Immune checkpoint inhibitors (ICIs) are anti-cancer agents with a completely novel mechanism that is different from those of conventional cytotoxic, hormonal, or molecularly targeted agents. In the late 2010s, anti-programmed cell death (PD-1) antibodies such as nivolumab [25,26] and pembrolizumab [27] were introduced into the therapy of head and neck carcinomas, including NPC. However, the efficacy of those agents against NPC is limited, with an overall objective response rate (ORR) of around 20% [28]. The efficacy is extremely low against tumors with low expression of programmed cell death ligand 1 (PD-L1), a ligand of PD-1 [27,28]. These facts indicate that novel therapeutic modalities are required in the treatment of NPC.

Farnesylation is a post-translational modification that attaches a lipid structure called a farnesyl group to the C-terminus of proteins [29,30,31]. Initially, farnesylation was investigated as a process to activate the RAS oncoprotein [32,33], and it was thus considered a therapeutic target in anti-RAS cancer therapy. Several clinical trials were conducted, with relatively promising results [34,35,36,37], but the results were not as favorable as expected. However, it was recently reported that an inhibitor of farnesylation showed promising results against H-RAS mutated head and neck carcinoma, both in cell lines and xenograft models and in clinical trials [38,39]. In addition, molecular evidence of the farnesylation of proteins other than RAS and its association with carcinogenesis has also been discovered [40,41]. Several reports demonstrate an association of farnesylation and NPC [42,43]. These reports suggest that protein farnesylation is also a potential therapeutic target in NPC.

In this review, we provide an overview of current reports on the molecular functions of farnesylated proteins, not limited to the RAS oncoprotein, in the context of NPC carcinogenesis and the potential use of anti-farnesylation therapy for NPC.

## 2. RAS Oncoproteins and Farnesylation

The *ras* genes were initially discovered as oncogenes of rodent retroviruses. Later, the same *ras* oncogenes were identified in human cancers, and a mutated *ras* oncogene activates its transforming property in several varieties of human cancers [44,45]. These findings suggest that the mutation of a *ras* oncogene is one of the key steps of carcinogenesis in human cancers. There are three mammalian *ras* genes: the H-*ras*, N-*ras*, and K-*ras* genes [46]. These three *ras* genes yield four RAS proteins: H-RAS, N-RAS, K-RAS4A, and K-RAS4B. The alternative splicing of the fourth exon of the K-*ras* gene produces two K-RAS proteins [47]. In a specific tumor, *ras* gene mutations are generally limited to only one of the three genes [48]. Which RAS isoform is mutated depends on the tissue and tumor type [49,50]. RAS proteins belong to the G protein family, members of which bind guanosine 5′-diphosphate (GDP) or guanosine 5′-triphosphate (GTP) [51,52,53]. RAS proteins are bound to GTP in exchange for GDP when activated [49,54]. Activated RAS activates multiple signaling pathways, including the Raf-MEK-ERK, PI3-K, Nore1, PLCε, Tiam1, and Ral pathways [55,56]. These pathways are implicated in RAS-mediated carcinogenesis.

RAS proteins must be bound to the cellular membrane to transduce extracellular signals and to exert transformational activity [57,58]. Farnesylation of RAS is reported to be a crucial mechanism for its association to membrane structures [32,33,59,60]. Furthermore, the inhibition of farnesylation was shown to have anti-tumor activity in mouse models [61,62]. Since then, blocking farnesylation has been recognized as a potential target for cancer therapy through inhibition of RAS activity [30].

## 3. Farnesylation, Its Biosynthesis, and Membrane Association by Farnesylation

Farnesylation was originally identified as an activation process of RAS proteins [32,33]. It is a lipid modification that attaches a farnesyl group to the thiol group of the cysteine residue of the CAAX motif (in which “C” is cysteine, “A” is aliphatic amino acid, and “X” is usually serine, methionine, glutamine, alanine, or threonine) in the C-terminus of a protein [29,30,31]. The farnesyl group is transferred to the protein by farnesyl transferase (FTase) and covalently bound to the cysteine residue of the CAAX motif. The AAX sequence is then removed though proteolysis by RAS-converting enzyme 1 (Rce1). Finally, the now-C-terminal cysteine is α-carboxymethylated by isoprenylcysteine carboxy methyltransferase (Icmt) [63]. The attached farnesyl group serves as a hydrophobic “tail” with high affinity to lipid bilayer structures. Thus, farnesylation is an essential process to mediate both protein–protein and membrane–protein interactions (Figure 1) [40,41].

Farnesyl pyrophosphate is a 15-carbon lipid that is synthesized via the mevalonate pathway. Acetyl coenzyme A (Acetyl-CoA) is converted to mevalonate and then to farnesyl pyrophosphate [41]. The mevalonate pathway is well known for the biosynthesis of cholesterol, and farnesyl pyrophosphate itself is a precursor of cholesterol [64]. This means that farnesyl pyrophosphate and cholesterol share a mechanism of biosynthesis, and the synthesis of farnesyl pyrophosphate is regulated by 3-hydroxy-3-methylglutaryl coenzyme A reductase (HMG-CoA reductase). HMG-CoA reductase is the rate-limiting enzyme of the mevalonate pathway and is regulated by sterol-mediated feedback or feedback from nonsterol isoprenoids, including farnesyl pyrophosphate [64].

Initial investigations into farnesylation targeted on the localization of RAS to the plasma membrane [63,65]. However, the localization of farnesylated protein is not limited to the plasma membrane; it also includes other membrane structures such as intracellular membrane organelles [65,66]. Lamin A is a protein required for nuclear envelope architecture and nuclear function [40]. In the process of maturation of lamin A, its precursor, prelamin A, localizes to the nuclear lamina through farnesylation [67]. PRL-1, -2, and -3 are protein phosphatases. They are localized to early endosome through farnesylation [68]. Additional examples are the centromeric proteins CENP-E and CENP-F. They are modified via farnesylation and participate in cell cycle progression [69,70]. CENP-F is localized to the nuclear envelope [71]. Ubiquitin C-terminal hydrolase-L1 (UCH-L1) is farnesylated and localized to endoplasmic reticulum membranes [72].

Farnesylated proteins attach not only to intracellular membrane structures but also to extracellular membrane structures. It was reported that farnesylation of RAS is required for the loading of activated RAS to exosome-like nanovesicles in a glioblastoma model [73]. A report on NPC showed that the farnesylation of UCH-L1 is a key process in the biogenesis of exosomes containing the viral oncoprotein LMP1 [43]. These reports suggest that farnesylated proteins are also associated with extracellular membrane structures.

## 4. Proteins Modified by Farnesylation Other Than RAS

RAS is not the only protein modified by farnesylation. More than 20 mammalian proteins are potentially farnesylated [41,74,75]. Several human proteins known to be farnesylated are listed in Table 1 [40,47].

One example of a farnesylated protein other than RAS is RhoB, which is a small GTPase involved in regulation of the actin cytoskeleton, cell motility, and proliferation [76,77]. Other examples are CENP-E and CENP-F, which are farnesylated at the C-terminal CAAX motif [70]. These proteins are centromeric proteins that function in the mitotic spindle checkpoint of the cell cycle [69].

In addition, malfunction of the farnesylation process of these proteins may cause diseases other than cancer. One good example of this occurs with lamin A. In the post-translational process of lamin A, farnesylation of its precursor, prelamin A, is mandatory. Malfunction of this process is closely related to the pathogenesis of progeria [78].

## 5. Development of Farnesyl Transferase Inhibitors

As a consequence of advances in research, farnesylation has come to be expected as a potential therapeutic target, not only against RAS, but also against other proteins. This expectation led to the development of several molecules that can act as farnesyl transferase inhibitors (FTIs). Those FTIs are classified into several groups.

The first group is the peptidomimetic inhibitors. These inhibitors are analogs of the CAAX motif and act as competitors for farnesyl transferase binding to the CAAX motif of the target protein. Initially, these compounds had poor membrane permeability and were unstable, but they were subsequently refined, and favorable bioavailability and potency were achieved [30,75].

The second class of FTIs comprises analogs of the farnesyl group. This class of FTIs has selective inhibiting activity in vitro but does not have relevant anti-tumor activity in vivo in animal models [79].

The third class is bisubstrate inhibitors, comprising complexes of analogs of the farnesyl group and CAAX motif. These inhibitors are highly potent, with anti-tumor activity observed in vitro and in vivo [80].

Another class is FTIs developed by the screening of drug libraries. For example, SCH66336, a tricylic inhibitor, and R115777, a nonpeptidomimetic inhibitor, were identified using this approach [79].

## 6. Nasopharyngeal Carcinoma and RAS

Activated *ras* genes have been detected in various human cell lines and in patient tumor tissues. Mutations of *ras* genes are found frequently in several cancers, such as colorectal [81,82], lung [83], and pancreatic [84,85] carcinomas. NPC reports indicate that mutation of a *ras* gene is quite rare, occurring in at most 1% of cases [86,87]. Another report resented the results of a higher proportion of NPC samples harboring *ras* mutations, around 20% in K-*ras* [88], although this is still much lower than the rates in pancreatic carcinoma (around 90%) [89] and in colorectal carcinoma (around 40%) [90]. In another EBV-related malignancy, NK/T lymphoma, it is also reported that *ras* mutation is relatively rare [91].

As noted above, LMP1 is a primary oncoprotein on EBV carcinogenesis [6]. On clinical samples of NPC, LMP1 expression is positively associated with metastasis of NPC [92]. How does LMP1 act on the RAS pathway? One report showed that LMP1 mediates ERK activation via a RAS-dependent pathway [13]. ERK is an MAPK activated in the Raf–MEK–ERK pathway, which is a downstream effector of RAS [44,45]. However, another report demonstrated that LMP1 regulates the Raf–MEK–ERK pathway in a RAS-independent manner [93]. This discrepancy might be explained by another report that described K-RAS as being suppressed by miR-1, which is regulated by LMP1 [94]. In other words, LMP1 indirectly activates RAS through the inhibition of miR-1. From these findings, it is probable that LMP1 is not a direct downstream effector of RAS, but might affect it in an indirect manner (Figure 2).

RAS mutation is relatively rare in NPC compared to other cancers, and the EBV oncoprotein LMP1 does not promote RAS, at least not in a direct manner. These findings on NPC and RAS seem to suggest that RAS-targeted therapy, including the inhibition of FTase, is useless against NPC. However, there is still potential in anti-RAS therapy. RAS is reported to be associated with resistance to chemotherapy and radiotherapy (Figure 3). It is widely accepted that chemotherapy and radiotherapy are important modalities of NPC therapy [95,96,97]. Although efficacy of salvage surgery for locally recurrent NPC is reported [98], indication for surgery is limited due to its anatomical features, so resistance to chemotherapy and radiotherapy leads directly to poor prognosis. In fact, it is reported that adjuvant chemotherapy against NPC patients with high risk of residual tumor did not improve prognosis [99]. The report shows that NPC resistant to initial therapy also tends to be refractory to following therapies. This means that measures to avoid the chemo-/radio- resistance are needed.

First, RAS is considered a promotor of anti-cancer-drug resistance. One well-known example is the anti-EGFR antibody and colorectal cancer. Colorectal cancers that harbor a mutation in K-*ras* or N-*ras* are unlikely to benefit from anti-EGFR antibodies such as cetuximab or panitumumab [100]. Therefore, evaluating K-*ras* mutation status is virtually mandatory when administering anti-EGFR antibody agents to colorectal cancer patients. There are several reports that RAS and its downstream effectors induce anti-cancer drug resistance in NPC. Reports suggest that the RAS pathway is related to cisplatin [101,102] or cetuximab [103] resistance in NPC. Both cisplatin [95,96] and cetuximab [104] are key anti-cancer agents used in the treatment of NPC. These findings suggest that even if inhibiting the RAS pathway does not exert direct anti-cancer activity, it might help in avoiding drug resistance. In fact, it has been reported that isoprenylcysteine carboxymethyltransferase (Icmt), the enzyme that contributes to the final step of farnesylation, is associated with RAS activation and chemoresistance in NPC [105].

Second, the activation of RAS and its downstream pathway influences radiation resistance. Several studies have shown that the overexpression of RAS proteins induce radiation resistance [106,107]. In human tumor cell lines, activating mutations of RAS isoforms have been shown to be closely related to intrinsic radiation resistance [108,109,110]. It has also been reported that the RAS downstream pathway is related to radiation resistance. The Raf–MEK–ERK pathway was implicated in radiation sensitivity. Inhibition of Raf by antisense c-raf-1 was reported to sensitize radiation resistant human squamous cell carcinoma cells to irradiation [111]. Additionally, a prostate carcinoma cell line expressing wild-type *ras* was found to be more sensitive to irradiation after MEK inhibition [112]. However, an MEK inhibitor, PD98059, could not sensitize two bladder cancer cell lines to irradiation [113]. These reports suggest that the activity of the Raf–MEK–ERK pathway on radiation resistance might be tissue or organ specific. In NPC, it was reported that the expression of Raf kinase inhibitory protein (RKIP) altered the radiosensitivity of NPC cell lines by mediating the Raf–MEK–ERK pathway. The authors demonstrated that overexpression of RKIP sensitized NPC cell lines to radiation-induced cell death, and underexpression of RKIP protected cells from radiation-induced cell death. They also noted that the expression status of RKIP in patients correlated with clinical features and outcome [114]. Another downstream pathway of RAS that is likely to mediate radiation sensitivity is the PI3K pathway. In one study, the induction of active PI3K resulted in increased radiation resistance. When a PI3K inhibitor was administered, this induction of radiation resistance was blocked [115]. Research in head and neck carcinoma showed that the expression status of PI3K on a clinical specimen correlates with the local control rate [116]. Additionally, in NPC, it has been reported that the PI3K pathway is associated with radiation resistance. In one report, a combination of inhibitors of PI3K and its downstream effector mTOR increased the radiosensitivity of NPC cell lines. The authors also demonstrated the effect of these inhibitors as radio-sensitizers in an NPC xenograft model [117]. Another group reported that leucine zipper tumor suppressor 2 (LZTS2) inhibited activation of the PI3K pathway and suppressed radiation resistance. They also reported that patients with low expression of LZTS2 had poor prognoses [118]. There is also a report focused on hypoxia. Hypoxic cells are resistant to irradiation. The authors reported that xenografts from cell lines with mutations in H-*ras* had markedly improved oxygenation by FTI treatment. In contrast, hypoxia did not improve in xenografts derived from cell lines without H-*ras* mutation [119].

## 7. Farnesylation and Nasopharyngeal Carcinoma

Even though efforts have been made to investigate farnesylation in the context of human malignancies, reports on farnesylation in NPC and other EBV-associated malignancies are quite limited. This is probably because efforts have been concentrated on farnesylation of RAS and malignancies with a high incidence of *ras* gene mutation, such as pancreatic carcinoma or colon carcinoma.

There is a report indicating that a combination of FTI and doxorubicin induces apoptosis through caspase-dependent early cleavage of the TRAF-1 [42]. In the report, the farnesyl transferase inhibitor BIM 2001 had only limited toxicity on cells derived from xenografts, C15 cells and C666-1 cells, but the cytotoxic effect of doxorubicin against C15 and C666-1 was enhanced when combined with BIM 2001.

However, there are several limitations to the report. First, no enhancement of cisplatin and bleomycin toxicity against both C15 and C666-1 cells was observed when combined with the same concentration range of BIM 2001. Second, the sensitivity of C666-1 cells to BIM 2001 was lower than that of C15 cells. A higher concentration of BIM 2001 was required by C666-1 than by C15 to enhance the toxicity of doxorubicin. In addition, incubation with the doxorubicin/BIM 2001 combination induced the apoptosis of C15 cells, but not that of C666-1 cells. These findings suggest that the effect of FTIs against NPC can vary from patient to patient. Third, the molecular target of BIM 2001 is unclear. It is reported that the RAS pathway induces cisplatin resistance, as noted above [101,102]. It is probable that the farnesylation of proteins other than RAS participates in the mechanism.

We reported that the C-terminal farnesylation of ubiquitin C-terminal hydrolase-L1 (UCH-L1) is associated with the transport of LMP1 to exosomes [43]. Exosomes are extracellular membrane vesicles 30 to 180 nm in diameter. Proteins, DNA, RNA, and lipids are loaded on exosomes. [120,121,122] Those “cargos” are transferred not only to neighboring cells, but also to distant organs through body fluids [123,124]. With these features, exosomes have been revealed to be important mediators of cell-to-cell communications in malignancies [125]. It is reported that LMP1 is secreted to exosomes produced in EBV- or LMP1-positive cells [125,126] and that EBV modulates the tumor microenvironment through secretion of viral proteins such as LMP1 to exosomes [127]. These LMP1-positive exosomes were reported to promote epithelial–mesenchymal transition (EMT) and facilitate the migration and invasion of donor EBV-negative cells [128]. Additionally, LMP1 increases the secretion of well-established oncogenic factors, such as FGF-2 [125] and HIF-1α [128]. Our report focused on the biogenesis of LMP1-positive exosomes and the mechanism of how LMP1 is loaded onto exosomes. We demonstrated that LMP1 is physically associated with UCH-L1 and that LMP1 is loaded onto exosomes through the process of farnesylation of UCH-L1. Moreover, farnesyl transferase inhibitor FTI-277 reduces the cellular migration and anchorage-independent growth of EBV-positive cell lines (Figure 4) [43]. However, there are also several limitations to the report. First, the status of UCH-L1 in the final product of LMP1-positive exosomes is unclear. In other words, it is not clear whether farnesylated UCH-L1 forms a complex with LMP1 on the LMP1-positive exosomal membrane or forms transient complexes only in the process of exosomal biogenesis and dissociates in the final product of LMP1-positive exosomes. Second, it was unclear whether UCH-L1 was the only protein affected by the inhibition of farnesylation in the experiment. As noted above, many proteins are affected by farnesylation, and the inhibitory effect of FTI-277 is not specific to UCH-L1. Thus, the observed inhibition of the transport of LMP1 to exosomes by FTI-277 could be mediated through the farnesylation of a protein other than UCH-L1. In the experiment, we utilized a C-terminal mutant plasmid of UCH-L1 in which farnesylation is impaired. This means that the chief mechanism should be mediated through the farnesylation of UCH-L1.

Additionally, research in breast cancer gives us useful suggestions towards the possible effect of FTIs in NPC. An FTI, tipifarnib, suppressed the expression of HIF-1α and Snail at a clinically relevant low dose [129]. HIF-1α and Snail are prometastatic transcription factors induced by EBV [125,130,131], and tipifarnib showed a favorable result in clinical trial on head and neck carcinomas [23]. Despite several limitations, these reports show that protein farnesylation has several roles in the molecular pathogenesis of NPC and is a potential therapeutic target.

## 8. Farnesylation as a Therapeutic Target

The fact that farnesylation is crucial for the membrane association and activation of RAS has prompted a huge effort to develop anti-farnesylation therapy. Farnesyl pyrophosphate is synthesized via the mevalonate pathway, and HMG-CoA reductase is a rate-limiting enzyme [41]. “Statins” such as lovastatin or pravastatin are inhibitors of the enzyme and are currently in clinical use for lowering cholesterol. Therefore, an initial investigation was conducted to utilize these agents for anti-farnesylation therapy. However, it was revealed that blocking RAS farnesylation requires a much higher concentration of lovastatin than the clinically effective concentration used for lowering cholesterol [132]. These findings suggest that utilizing the HMG-CoA reductase inhibitors, including statins, as RAS-inhibiting anti-cancer agents would be ineffective.

Later, FTIs were initiated as a class of experimental cancer drugs. Several clinical trials have been conducted on FTIs and relatively favorable results have been obtained in the early stages [34,35,36]. However, the overall results were less favorable than anticipated. A trial on juvenile myelomonocytic leukemia obtained a favorable initial response rate without increasing toxicity but failed to reduce relapse rates or improve long-term overall survival [37]. The most disappointing result was obtained in a trial against advanced pancreatic cancer in which most cases carried K-*ras* mutant [133].

There are several reasons why FTIs do not work against K-*ras* mutated cancers. First, K-RAS4B has much higher affinity to FTases than other isoforms of the RAS protein, which makes it more difficult for FTIs to deactivate [134,135]. Second, RAS activation by farnesylation is bypassed by another isoprenylation process, geranylgranylation. Under physiological conditions, the post-translational modification of RAS proteins is predominantly farnesylation catalyzed by FTase. However, geranylgeranylation catalyzed by geranylgeranyl transferase-1 activates K-RAS and N-RAS under FTI treatment [136,137].

Despite these disappointing results, there is still good news. H-RAS is predominantly activated through farnesylation even under FTIs [136]. Fortunately, head and neck carcinoma harbors relatively a high incidence rate of mutated H-*ras* gene compared to other cancers. The COSMIC database (https://cancer.sanger.ac.uk/cosmic, accessed on 20 March 2022) reports that H-RAS is mutated in 6% of head and neck carcinoma, compared to 1% in breast and colorectal carcinoma, 0.6% in lung carcinoma, and 0.1% in pancreas carcinoma.

A report demonstrated the efficacy of an FTI, tipifarnib, against head and neck squamous cell carcinomas (HNSCCs) in cell lines and xenograft models [38]. Tipifarnib treatment displaced both mutant and wild-type H-RAS from the membrane structure in these cell lines. However, proliferation, mortality, and spheroid formation were inhibited only in H-RAS mutant cell lines. Anti-tumor activity was also assessed in patient-derived xenograft (PDX) models. H-RAS mutated xenografts were highly sensitive to tipifarnib, and tumor regression was induced. In contrast, the H-RAS wild-type xenografts proliferated even under tipifarnib treatment. Recently, the results of a phase Ⅱ trial of tipifarnib monotherapy against H-RAS mutated HNSCCs were published. The outcome was favorable, with a 55% objective response rate and tolerable adverse events. The median progression-free survival was 5.6 months, compared to 3.6 months on the last prior therapy, and the median overall survival was 15.4 months [39]. This result is quite promising, and several other clinical trials against HNSCCs are ongoing. Table 2 is the list of clinical trials of FTIs against HNSCCs including NPC.

However, there is a limitation in the application to NPC therapy. According to the COSMIC database, only 1% of NPC cases harbor mutated H-RAS. In fact, there was no clinical trial of FTIs dedicated to NPC patients as far as we explored on ClinicalTrials.gov (https://clinicaltrials.gov/, accessed on 29 May 2022). Although it seems that treatment of NPC with FTIs is ineffective, there is still hope. FTIs might be effective even in the absence of mutated H-RAS through inhibiting the functions of proteins other than RAS. In fact, although this is an in vitro study, even cell lines that do not harbor RAS mutations are sensitive to FTIs [47,50,75]. In addition, FTIs might be effective as chemo- or radio- sensitizer. An important feature of FTIs is that they have very low toxicity for normal cells at a concentration sufficient to inhibit the growth of transformed cells [47]. Even if FTIs are not effective enough for use as a monotherapy, the low toxicity of FTIs means that they can easily be utilized in combination with other therapeutic measures. In fact, there is an ongoing clinical trial of a combination of tipifarnib and alpelisib, a PI3K inhibitor. The trial targeted recurrent/metastatic HNSCCs, and the patients were not limited to those with increased H-RAS dependency (Table 2).

## 9. Conclusions

Research on farnesylation was initially focused on the RAS activation process, and several classes of FTIs were developed. More and more molecular evidence then showed that proteins other than RAS are also farnesylated, including several oncogenic proteins. The results of these in vitro and in vivo experiments encouraged clinical trials on FTIs. Several clinical trials obtained at least partially favorable results, but the overall results were disappointing.

However, advancements in cancer genomics have opened new paths. Head and neck carcinomas have a relatively high incidence rate of H-*ras* mutation, which became an ideal target of FTIs. Promising results from the application of FTIs to head and neck carcinomas both in laboratory experiments and clinical trials have been published. To date, they have focused on H-*ras* mutated tumors, and application to NPC is limited. However, it has been shown that several proteins other than RAS participate in EBV oncogenesis through farnesylation. These findings show the potential of FITs in NPC therapy. Further advancements and breakthroughs in the research and clinical applications of FTIs for NPC therapy are awaited.

## Figures and Tables

**Figure 1 cancers-14-02826-f001:**
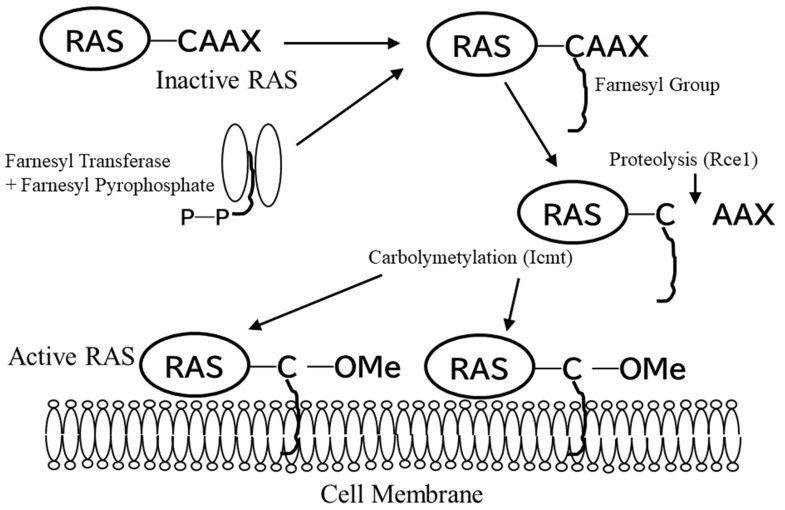
Schematics of farnesylation and membrane association of RAS oncoprotein. RAS processing and membrane association is critical for its transforming activity. C-terminus of RAS protein is first fanesylated followed by proteolysis, carbocxymetylation. C = cysteine, A = aliphatic amino acid, X = any amino acid, Icmt, isoprenylcysteine carboxymetyltransferase; Me, metyl group; P-P, pyrophosphate group; Rce1, RAS-converting enzyme 1.

**Figure 2 cancers-14-02826-f002:**
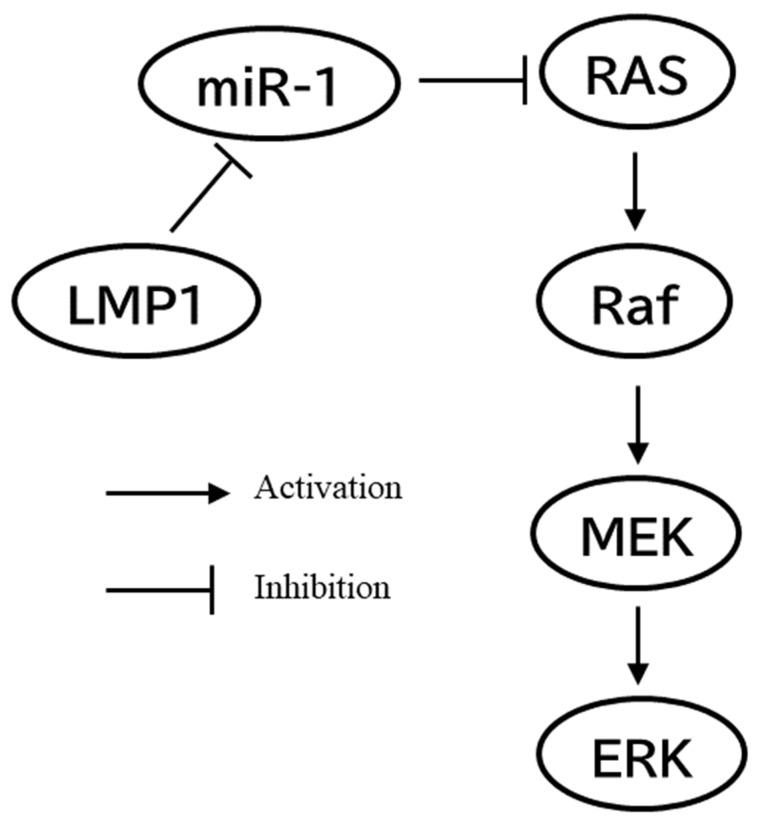
Proposed activation process of RAS pathway by EBV oncoprotein LMP1. RAS is not a downstream effector of LMP1; however, they are indirectly associated. LMP1 activated downstream effector of RAS, Raf-MEK-ERK pathway. Additionally, LMP1 activate RAS through inhibition of miR-1.

**Figure 3 cancers-14-02826-f003:**
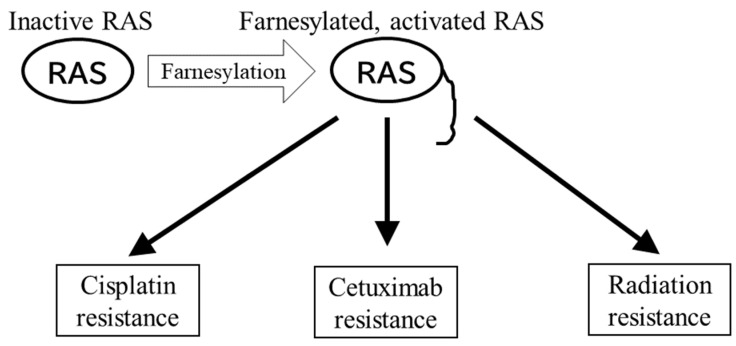
Farnesylation in drug resistance of NPC. RAS is activated through farnesylation. Farnesylated RAS promotes drug resistance and radiation resistance.

**Figure 4 cancers-14-02826-f004:**
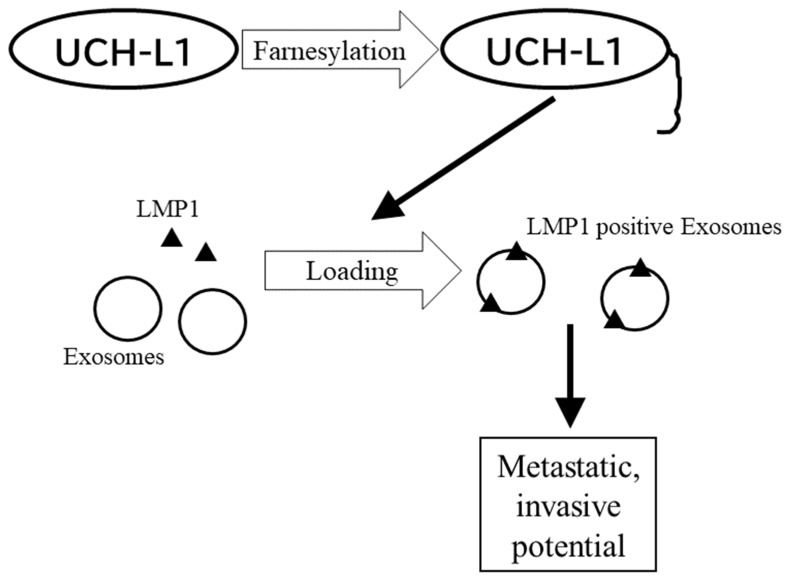
Farnesylation in metastasis and invasion of NPC in drug resistance of NPC. Fanesytaion of UCH-L1 plays a role in loading of LMP1 to exosomes. LMP1-positive exosomes promote metastatic, invasive potential of NPC.

**Table 1 cancers-14-02826-t001:** Farnesylated proteins.

Proteins	Functions
H-, K-, N-RAS	GTPase, signal transduction
Rho	GTPase, signal transduction
Rheb	GTPase, signal transduction
PRL family	Tyrosine phosphatase
CENP-E	Kinesin motor protein
CENP-F	Chromosome passenger
HDJ	Cochaperone
Nuclar lamins	Nuclear envelope protein
UCH-L1	De-ubiquitinating enzyme

**Table 2 cancers-14-02826-t002:** List of clinical trials of FTIs against HNSCCs R/M, recurrent or metastatic.

IDs on ClinicalTrials.gov (accessed on 29 May 2022)	Phase	Agents	Tumor Types	References
NCT02383927	Phase 2	Tipifarnib	HNSCCs with *H-ras* mutations	[39]
NCT03719690	Phase 2	Tipifarnib	HNSCCs with *H-ras* mutations	Ongoing trial
NCT04997902	Phase 1/2	Tipifarnib with Alpelisib	*PI3KCA* or *H-ras*-dependent R/M HNSCCs	Ongoing trial

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
