# Peer review of "Protein Farnesylation on Nasopharyngeal Carcinoma, Molecular Background and Its Potential as a Therapeutic Target"

_cancers, 2022, doi:10.3390/cancers14122826_

Round 1

Reviewer 1 Report

Manuscript “Farnesylation and Nasopharyngeal Carcinoma, Ras and other “  covers and interesting topic, and gives relatively new way of looking at anti-Ras therapy. However, it is written as a separate chapters about EBV in NFC, clinical features of NFC, explanation of farnesylation and description of ras proteins. It would be beneficial to rearrange the text in a way that those chapters are more clearly connected and that focus really is on NFC and farnesylation, and not, as it seems now, Ras and similar proteins in cancer. Authors should also avoid repetition of explanations given at some point in the text in following chapters. The title should be more meaningful and in accordance to text, and the strong conclusions such as EBV is a cause for NFC metastasis (it is involved and could even trigger it, but cause is a bit to defined given the lack of knowledge about various aspect of EBV in metastatic NFC) or at the beginning of the chapter 4. „These findings suggest that the farnesylation of proteins other than Ras is potentially related to carcinogenesis.” should be avoided. Table one should define if listed proteins are in humans or all mammals, and what was the criteria for selecting those few for the list. Nomenclature for proteins in different animals should be taken into account (e. g. RAS in humans, Ras in mice, etc.)

Reviewer 2 Report

Kobayashi et al. made a review article, in which they try to describe the relationship between the farnesylation proteins and the tumorigenesis of nasopharyngeal carcinoma (NPC). The critical issue in this topic is that ras mutations are rarely found in NPC, as well as in other EBV-associated malignancies. However, farnesylation is found to be important in drug resistance and metastasis of NPC. This review gives a detailed introduction on the backgrounds of NPC, Ras, and farnesylation. The authors also give abundant references to introduce the potent application of anti-farnesylation in NPC therapy. Some minor comments are listed as follows.

1.     Please list the clinical trials on the table regarding NPC therapy using anti-farnesylation strategy.

2.     Please draw simple charts to describe the roles of farnesylation in drug resistance and metastasis of NPC.

3.     Farnesylation in NPC tumorigenesis may include EBV(LMP-1)-dependant and independent mechanisms. Is there any pathological difference between them?

Reviewer 3 Report

This is a well-written and comprehensive narrative review, shedding light on the role of farnesilation within the Ras oncogene pathway, with a special reference to the nasaopharyngeal cancer setting.

Few issue in this manuscript deserve an improvement:

1. The title is confusing, and should be rewritten to better synthesize the aim of the study

2. Paragraph 6 "Nasopharyngeal carcinoma and Ras" contains a wide section regarding the interplay between LMP1 and Ras pathway in settings not directly related with NPC. I would start the paragraph with the description of evidence regarding NPC and then discuss them in light of evidences based on other settings.

3.  I would focus more on the impact of chemo/radio resistance in the specific setting of NPC treatment, thus giving a stronger support to the  statements regarding a possible role of Ras-targeted therapy in NPCs as a chemo-radio-sensitizer.

4. Lines 230-231: Authors state that "NPC is not suitable for surgery due to its anatomical features, so resistance to chemotherapy and radiotherapy leads directly to poor prognosis". This statement should be re-considered in light of the outcomes of endoscopic salvage Nasopharyngectomy (see for example doi: 10.1016/S1470-2045(20)30673-2.; doi: 10.1186/s40463-020-00482-x; doi: 10.3389/fonc.2021.716729 etc.)

Round 2

Reviewer 3 Report

Authors sufficiently responded to the previously risen issues, and the manuscript has significantly improved.

I'd just recommend the manuscript to be proof read for English grammar and style, to enhance readability.